# International Tourism Development in the Context of Increasing Globalization Risks: On the Example of Ukraine's Integration into the Global Tourism Industry

**Yurii Kyrylov [1], Viktoriia Hranovska [1], Viktoriia Boiko [1] ⬤, Aleksy Kwilinski [2]⬤ and Liudmyla Boiko [1,*]⬤**

[1] Department of Economics, Kherson State Agrarian University, 23 Stritenska Street, 73006 Kherson, Ukraine; kirilov_ye@ukr.net (Y.K.); vgranovska@ukr.net (V.H.); boiko.vo17@gmail.com (V.B.)

[2] The London Academy of Science and Business, 120 Baker Street, London W1U 6TU, UK; a.kwilinski@london-asb.co.uk

\* Correspondence: boiko.mila7@gmail.com

**Abstract:** Today, international tourism is one of the most affected sectors of the economy due to the global COVID-19 pandemic. The main purpose of this article is to analyze current trends and identify prospects for the international tourism development in the context of increasing globalization risks in the world, using the example of Ukraine's integration into the global tourism industry, as Ukraine is located in the centre of Europe and belongs to a number of countries with developing economies, and has the potential to expand its tourism industry, which may be of interest to the international scientific community in terms of overcoming the bifurcation point of its economic development. Analyzing the tourism industry, as one of the most progressive sectors of the world economy, we used general scientific and special research methods (abstract-logical, statistical, systemic analysis and synthesis, abstract-theoretical, and correlation-regression analysis). The paper analyzes major indexes of international tourism development in the modern globalized world and details the risks emerging during the global COVID-19 pandemic. It examines the global dynamics of tourism flows, where France, Spain, and the USA are unquestionable leaders. The study considers foreign exchange earnings of international tourism and the industry contribution to the gross domestic product of countries being an essential component of national budgets. Based on the study conducted, there were developed reliable forecast models for the tourism industry development in the countries under research. These models will provide an opportunity to generate reliable forecasts, which will allow timely identification of potential threats and making effective decisions to address them. At the same time, the issues of managing information support of economic entities in the field of international tourism need to be further developed in order to reduce risks.

**Keywords:** international tourism; globalization risks; global tourism trends; tourism industry; information technology; travel

## 1. Introduction

International tourism, as part of the process of global development and integration, has become one of the influential factors on which depend the economy growth, the increase in the country's competitiveness in world markets, and improvement of the population's welfare. Currently, the field of tourism employs more than 250 million people, i.e., every twelfth employee in the world. It accounts

for 7% of total investment, 11% of global consumer spending, 5% of all tax revenues and a third of world trade in services (UNWTO Statistics).

The global crisis caused by the coronavirus pandemic has radically changed the tourism industry around the world. And so there occurred a need to reconsider approaches to the usual work in the tourism industry. The Internet and digital transformation (Watkins et al. 2018; Ziyadin et al. 2019a, 2019b; Miskiewicz 2019, 2020; Miskiewicz and Wolniak 2020; Dzwigol et al. 2020) are predicted to become an important factor in providing information to consumers and marketing, as well as organizing the work of various tourism professionals. It can be assumed that the vast majority of improvements in organizing the tourism business will be based on organizing virtual solutions and remote work, as well as on educational programs, which in many countries will remain online after the COVID-19 crisis.

It is believed that the world still faces serious challenges and ordeals, starting with the indefinite duration of the pandemic period and ending with restrictions on movement, all these being in the context of the global economic recession. Countries around the world are implementing a wide range of measures to minimize the effects of the COVID-19 pandemic and stimulate the recovery of the tourism sector (Qiu et al. 2020). However, what will happen in reality depends on international financial institutions and powerful financial donors. This ensures relevance of the issue of developing the international tourism industry, and attaches scientific and practical significance to studying this problem in today's crisis conditions.

## 2. Literature Review

At the beginning of the 21st century, tourism turned into the global phenomenon of today and became a powerful component of economic development and a factor in shaping the image of many countries of the world (Hall 2019).

The problems and trends of international tourism development have been studied by a number of scientists. Namely, Ural (2016) considers the crisis phenomena and catastrophes that occur in the world, their consequences, as well as providing prerequisites for risk management processes for sustainable development of the tourism industry. The research results by Carballo et al. (2017) point to the existence of five types of risks that affect tourists when traveling internationally—health risk, crime risk, accident risk, environmental risk, and natural disaster risk—and that uncontrolled risks in tourism industries are perceived as more important than those controlled. Based on psychological theory and cognitive science about the brain, Wen (2018) in his work studies the mechanism of tourist risk perception using electroencephalography (EEG) experiments. A number of researchers (Althonayan and Andronache 2019; Fiol 2019; Abdurrahman et al. 2020; Biao et al. 2014) study the impact of enterprise risk management (ERM) on the results of their activities, while recommending the establishment of a committee (Malik et al. 2020) on risks at the board level (BLRC), an important governance mechanism controlled by the ERM.

Casidy and Wymer (2016) explore the moderating role of tourism risks in the relationship between satisfaction, loyalty and willingness to pay a high price for receiving quality services. Sofiichuk (2018) and Romanova (2020) identify and evaluate the existing risks of the tourism industry in Ukraine and offered recommendations on measures to be taken by the state to manage and prevent their occurrence. Williams and Baláž (2015) argue that risks in the tourism industry can be systematized and predicted.

At present, information technology (Cheunkamon et al. 2020; Cai et al. 2019; Natorina 2020) plays an important role in a globalized world. Firoiu and Croitoru (2015) focus on the impact of information technology on the development of enterprises providing services to the international tourism industry. Suyunchaliyeva et al. (2020) discuss possible challenges and current views on the link between information technology and tourism. Watkins et al. (2018) and Ziyadin et al. (2019a, 2019b) explore the impact of digital development on the tourism industry and the benefits of information technology in promoting e-tourism due to new technological conditions such as e-visa, e-tickets, online payments, advance booking of services and others. Modern information methods and technologies in

tourism, promoting brands of countries and regions, as well as individual tourism enterprises through Internet marketing and social networks are considered by Kozhukhivska et al. (2020).

Such scholars as Trusova et al. (2020) investigate the imperatives of developing the market of tourism services which determine the parameters of the aggregate value of the meso- and local levels' sub-indices in the spatial polarization of the regional tourism system. They substantiate the methodology of spatial polarization of the regional tourism system, which provides convergence of the infrastructural space of tourism services, formation of an innovative nucleus, minimization of destructive factors' manifestation, and balancing of interests of regions and the country as a whole. Hranovska et al. (2020) note different approaches to interpreting professional competence in tourism.

In their studies, Boiko (2020) and Romanenko et al. (2020) consider rural green tourism capable of making a considerable contribution to economic growth and improving the quality and comfort of rural dwellers' lives as one of the components of international tourism. In the near future, Ukrainian villages will be capable of creating a system of the world-level tourism services without losing their originality and diversity of national cultures. Currently, urban dwellers comprise more than half of Ukraine's population, and it explains a great potential for forming demand for services of rural green tourism and for creating a competitive area for foreign tourists. Kyrylov et al. (2020) and Von Essen et al. (2020) argue that entrepreneurship in the field of green tourism is a specific type of tourism activity that has a social, environmental, and economic impact on the rural areas' development in the region.

Dalevska et al. (2019); Kharazishvili et al. (2019, 2020); and Kwilinski et al. (2020) offered a methodology and instruments for economic–mathematical modeling to evaluate the degree of the development of international commercial and investment relationships, the degree of a lifespan, living standards and prosperity of international structures, the development of international tourism under the influence of sources for economic growth. In his research, Dzwigol (2020) uses the model of fuzzy logic to determine the level of regions' investment attractiveness, taking into consideration the factors of "security of investment activities" (criminogenic, environmental, and political), that allow determining unquestionable leaders of a region. The latter two factors should be attributed to such indexes as tourism potential and self-awareness of the population. Using the "Fixed Effect" model with an individual effect, Waciko and Ismail (2019) evaluated a positive impact of international tourists' arrivals on the gross domestic product (GDP), international tourism revenues, and international tourism expenditures and also a negative impact on GDP, partly related to the total employment in tourism sector. Murshed et al. (2020) assessed the relationship between demand for international inbound tourism (IITD), regional trade integration and the transition to renewable energy sources (RET).

Stezhko et al. (2020) and Ziyadin et al. (2018) pay special attention to the impacts of globalization on tourism, considering both positive and negative effects of globalization processes on different areas of social life. Reshetnikova and Magomedov (2020) offer practical recommendations concerning the development of business activities in tourism that can be used in regional, national and international markets of business travels. Zhang and Zhang (2020) proved the positive impact of tourism on gender equality. In their research, Lukianenko et al. (2019) present their individual view on the impact of globalization, internalization, and transnationalization on developing international tourism industry, substantiating the specificity of institutional transformations, caused by scale, structural, and other changes in the global economy.

Such scholars as Tran et al. (2020) and Rodríguez-Antón and del Mar Alonso-Almeida (2020) dealt with the issues of developing international tourism under conditions of COVID-19 global pandemic. The crisis due to quarantine restrictions has forced the tourism business to step out of its comfort zone and look for innovative ways to develop and operate. To minimize personal contacts between the tour operator and the consumer of tourist services, it is proposed to introduce workplace automation technologies using service robots (Ivanov 2020; Ivanov and Webster 2020). Such automation transforms and creates new jobs in the industry.

The risk and uncertainty of statistical results in the tourism industry is critical in the decision-making process, as due to the lack of complete information it is impossible to accurately

predict certain events. To obtain scientifically sound data, you can use different methods of statistical analysis: for example, correlation analysis evaluates the relationship between two or more variables, and regression analysis is used to optimize and predict the selected parameter for the near future (Bewick et al. 2003; Zaid 2015; Shyti and Valera 2018; Calvello 2020).

Summarizing the scholars' views, we can state that international tourism is a form of international economic relations, affecting a considerable number of economic sectors, consisting of many social components and aimed at improving the countries' well-being.

However, the issues of international tourism development in the context of global risks still need further analysis, which determined the purpose of this study. The main purpose of the article is to analyze current trends and determine the prospects for international tourism in the world in the face of increasing globalization risks.

## 3. Materials and Methods

The research is based on the statistical data and methodology of international tourism accounting proposed by the UNWTO and used by the World Bank (n.d.). In particular, the inbound direction of international tourism is evaluated in such natural and cost indexes as: international tourist arrivals and income from international tourism. According to the UNWTO methodology, the number of arrivals is considered as the quantity of registered visitors of this or that country who are not its residents for a particular period of time. In this case, they cannot stay in a destination country for more than one year and be engaged in activities paid by local budgets. All of them can be divided into one-day visitors and tourists (visitors staying in a destination country for more than one night).

While calculating arrivals, international tourism prefers registration on the national borders. However, not all countries can collect such data. Therefore, other indexes can be used instead of these, in particular, registration at hotels and similar institutions. The UNWTO statistics of tourism income covers receipts (US$), received by a destination country from international tourism for a certain period of time, as a rule, for a year. They consist of expenditures of visitors arriving to a certain country. International tourism expenditures are spending by international outbound visitors in other countries including payments to foreign carriers for international transportation. These expenditures may include both the data on citizens leaving abroad and the data on one-day visitors, except those cases when they should be distinguished in an individual classification. The main items of foreign tourists' expenditures in a destination country are lodging, meals, domestic transport, and fuel, excursions, entertainments, shopping and others. This index covers income earned from both tourists and one-day visitors. Expenditures of the latter can be considerable, especially in those cases when they live in border territories and go to the neighboring countries to buy goods and services. Such trips can be regular, which makes them an important source of income from international tourism.

The forecast of the tourist market conjuncture provides the opportunity to define prospects of its development in the future and underlies planning the activity of the businesses in this sector. In economics, the most common form of connection among the parameters chosen for the research (international income from tourism) is a linear form: $y = a_0 + a_1X$, where $a_0$ and $a_1$ are unknown values; thanks to this function we can generate forecasts for the near future. The value of the approximation validity is shown by the coefficient of determination $R^2$, which is a number from 0 to 1; the trend line is most true to reality when the value is close to 1; otherwise you can superimpose several trend lines and select more accurate models in the class of simple elementary functions: polynomial, logarithm, or power.

Using modern tools of economic and mathematical modeling, we have constructed linear and polynomial models for forecasting international tourism revenues for the countries analyzed. For the purpose of visual representation of computational data, it is expedient to use a graphic method in the Excel editor: to construct the scattering diagram; impose a trend line, trend parameters are calculated by the method of least squares; derive the equation of the curve and the reliability coefficient of the approximation validity, to assess the proximity of the curve to real data (Zaid 2015).

## 4. Results

Globalization and technological progress have contributed to considerable growth of international tourism in the global economy where its share makes up about a tenth. This industry has recently been considered as its global driver since it developed much faster than the global economy on the whole, its income considerably exceeding the cost of exporting fuels and raw materials (UNWTO n.d.).

The analysis of tourism income indicators (Table 1) shows that the five leading countries in terms of international tourism revenues have not changed. The United States still occupies the first place, being a huge tourist market with a highly developed infrastructure, hotel chain, and transport industry. Thailand has become one of the most important tourist destinations, ranking among the five most developed countries in the world, especially after it began to develop new beaches on the southern coast of the country and organize cultural and educational trips to the northern regions. In 1960, the country was visited by about 80 thousand foreign tourists, and in 2019, their number reached 39.8 million, and tourism industry brought USD 66.2 billion to Thailand (2019), which account for a fifth of the national income. As far as Ukraine is concerned, it has huge potential in the field of tourism, showing a rather low growth rate of this sector—only USD 2 billion in 2019, which is 85th in the ranking of 186 countries (NationMaster (a)). However, the growth of revenues from the tourism industry in 2018 amounted to 12.4% and this is due to the influence, to some extent, of external factors. By developing inbound tourism, Ukraine is becoming increasingly popular with foreign tourists, and travelers are also attracted by the availability of environmentally friendly products and recreational areas, which is a significant competitive advantage. With the change of government in 2019, the existing system of international relations was unbalanced, which introduced uncertainty regarding the development of many sectors of the economy, including the tourism business, which led to a decrease in revenues by 12.1%.

**Table 1.** International Tourism Revenues (in billion US$).

| Country | Years | | | | | | In % 2019 to 2018 |
|---|---|---|---|---|---|---|---|
| | 2014 | 2015 | 2016 | 2017 | 2018 | 2019 | |
| The USA | 235.990 | 249.183 | 245.991 | 251.544 | 256.145 | 264.576 | 103.3 |
| Spain | 71.656 | 62.449 | 66.982 | 75.906 | 81.250 | 81.368 | 100.11 |
| France | 67.402 | 66.441 | 63.557 | 67.936 | 73.125 | 72.889 | 99.7 |
| Thailand | 38.451 | 44.881 | 48.459 | 57.057 | 65.242 | 66.156 | 101.4 |
| Germany | 58.721 | 50.669 | 52.229 | 56.330 | 60.260 | 60.254 | 100.0 |
| Italy | 45.562 | 41.415 | 42.423 | 46.719 | 51.602 | 50.895 | 98.6 |
| Great Britain | 51.582 | 50.904 | 47.777 | 47.719 | 48.515 | 49.580 | 102.2 |
| Australia | 35.736 | 36.249 | 39.059 | 43.975 | 47.327 | 48.085 | 101.6 |
| Japan | 20.790 | 27.285 | 33.456 | 36.978 | 45.276 | 45.523 | 100.5 |
| Hong Kong | 46.352 | 42.491 | 37.838 | 38.170 | 41.870 | 40.737 | 97.3 |
| Ukraine | 2.264 | 1.662 | 1.723 | 2.019 | 2.269 | 1.995 | 87.9 |

The source: Developed by the authors based on the NationMaster (a).

A logical continuation of our analysis is the calculation of the country's income from international tourism per capita (Table 2). Hong Kong has the highest rates of income from international tourism among the studied countries. In 2014, the country received USD 6.52 thousand per capita; during the analyzed period, this indicator fell by 17.3% and in 2019 amounted to USD 5.39 thousand. The reason for this decline is competition from neighboring Macau, which attracts tourists with its gambling establishments and large shopping centers. More than one thousand dollars of per capita income is in Australia, France, and Spain. In Ukraine, which is located in the heart of Europe and has all the conditions for proper economic development through tourism, this figure, for years of research, in average accounts for only USD 46.8 per person, which indicates a significant lag behind the world's leading countries in terms of tourism infrastructure and quality of tourist services.

**Table 2.** International Tourism Revenues per 1 person (in thousands of US$).

| Country | Years | | | | | | In % 2019 to 2018 |
|---|---|---|---|---|---|---|---|
| | **2014** | **2015** | **2016** | **2017** | **2018** | **2019** | |
| The USA | 0.74 | 0.78 | 0.76 | 0.77 | 0.78 | 0.80 | 102.6 |
| Spain | 1.50 | 1.35 | 1.44 | 1.63 | 1.74 | 1.74 | 100.0 |
| France | 1.02 | 1.03 | 0.99 | 1.05 | 1.13 | 1.12 | 99.1 |
| Thailand | 0.57 | 0.67 | 0.72 | 0.84 | 0.96 | 0.97 | 101.0 |
| Germany | 0.72 | 0.62 | 0.63 | 0.68 | 0.73 | 0.73 | 100.0 |
| Italy | 0.74 | 0.68 | 0.70 | 0.77 | 0.85 | 0.84 | 98.8 |
| Great Britain | 0.81 | 0.78 | 0.73 | 0.72 | 0.73 | 0.74 | 101.4 |
| Australia | 1.59 | 1.51 | 1.60 | 1.77 | 1.88 | 1.88 | 100.0 |
| Japan | 0.16 | 0.21 | 0.26 | 0.29 | 0.36 | 0.36 | 100.0 |
| Hong Kong | 6.52 | 5.81 | 5.13 | 5.15 | 5.59 | 5.39 | 96.4 |
| Ukraine | 0.051 | 0.039 | 0.041 | 0.048 | 0.054 | 0.048 | 88.89 |

The source: Developed by the authors based on the NationMaster (a). Ranking of Countries in the World by Population in 1980–2024 2019 (2019).

Due to global risks, there emerges a need to increase the level of control over economic processes in the world. Forecasting future events using economic and mathematical models can be an effective tool. To graphically represent the dynamics functions, we constructed trend curves on the correlation field. With the help of the linear and polynomial models obtained it is possible to forecast the change of the indicator for the following periods in the countries studied (Figures 1–6).

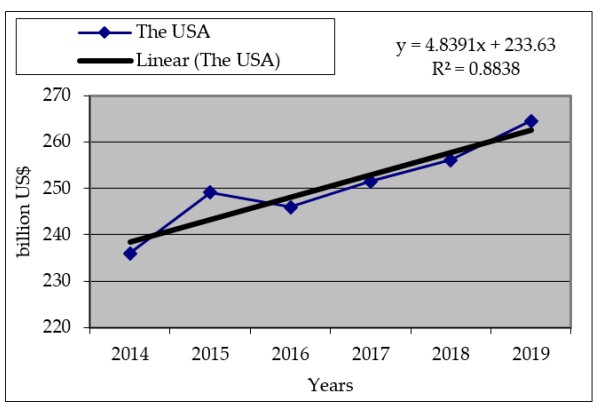
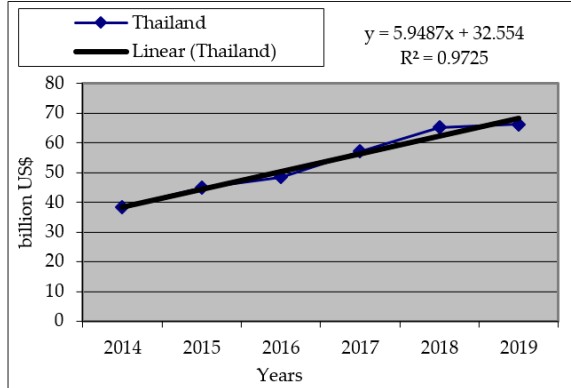

**Figure 1.** A Correlation Field and Linear Trend Curve of International Tourism Revenues for the USA and Thailand.

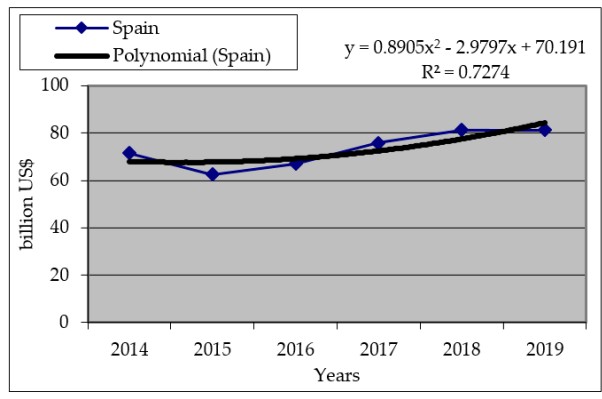
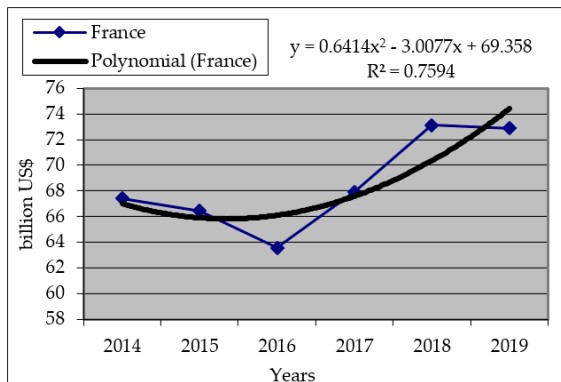

**Figure 2.** A Correlation Field and Polynomial Trend Curve of International Tourism Revenues for Spain and France.

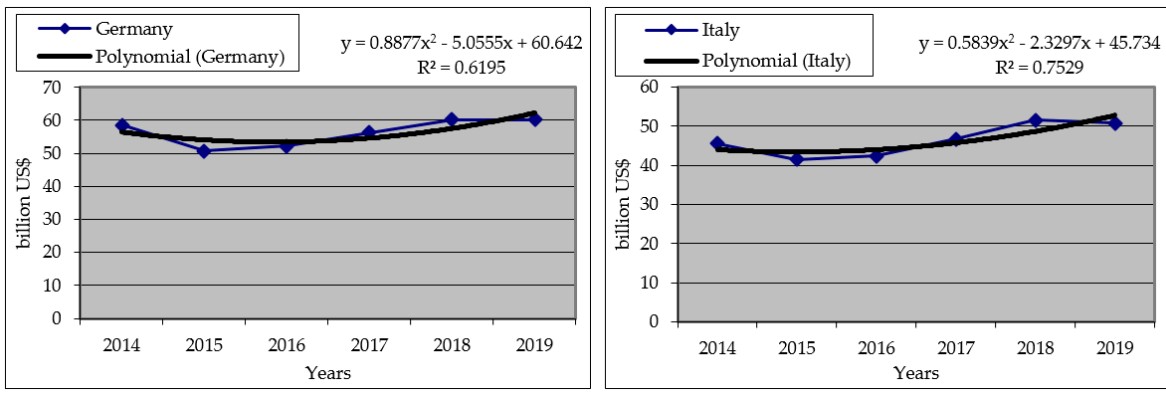

**Figure 3.** A Correlation Field and Polynomial Trend Curve of International Tourism Revenues for Germany and Italy.

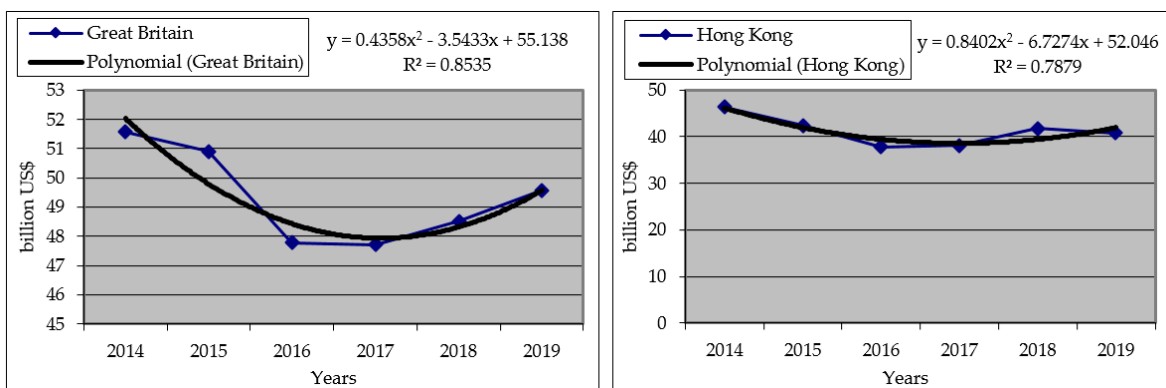

**Figure 4.** A Correlation Field and Polynomial Trend Curve of International Tourism Revenues for Great Britain and Hong Kong.

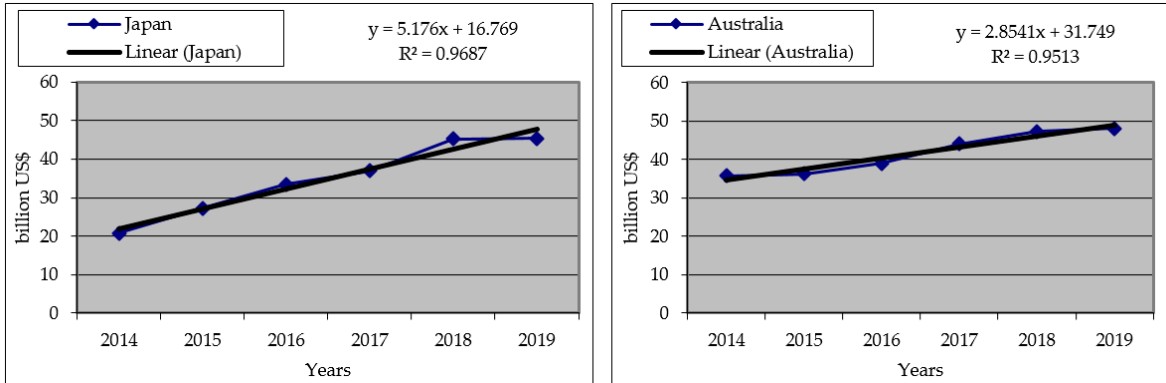

**Figure 5.** A Correlation Field and Linear Trend Curve of International Tourism Revenues for Japan and Australia.

Considering the graphical interpretation of the constructed models, it can be maintained that the points of the correlation field relative to the trend line are located dynamically and evenly, i.e., the calculations made are accurate, there are no significant errors, and we can use a reliable forecast model in future calculations. Aggregate indicators are summarized in Table 3.

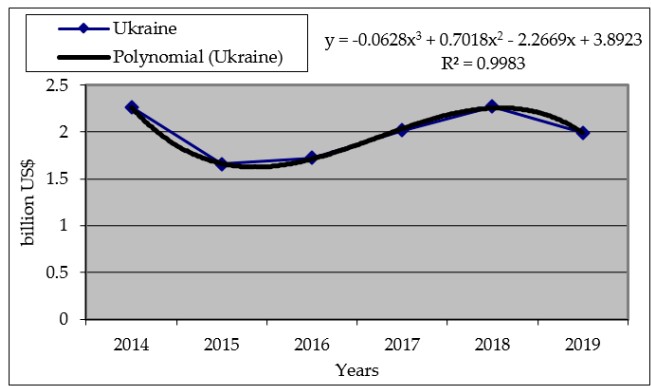

**Figure 6.** A Correlation Field and Polynomial Trend Curve of International Tourism Revenues for Ukraine.

**Table 3.** Dynamics Functions to Forecast International Tourism Revenues for the Countries Analyzed for the Near Future.

| Countries | Dynamics Function | $R^2$ | r |
|---|---|---|---|
| The USA | $y = 4.8391x + 233.63$ | 0.88 | 0.94 |
| Spain | $y = 0.8905x^2 - 2.9797x + 70.191$ | 0.73 | 0.85 |
| France | $y = 0.6414x^2 - 3.0077x + 69.358$ | 0.76 | 0.87 |
| Thailand | $y = 5.9487x + 32.554$ | 0.97 | 0.98 |
| Germany | $y = 0.8877x^2 - 5.0555x + 60.642$ | 0.62 | 0.79 |
| Italy | $y = 0.5839x^2 - 2.3297x + 45.734$ | 0.75 | 0.87 |
| Great Britain | $y = 0.4358x^2 - 3.5433x + 55.137$ | 0.85 | 0.92 |
| Australia | $y = 2.8541x + 31.749$ | 0.95 | 0.97 |
| Japan | $y = 5.176x + 16.769$ | 0.97 | 0.98 |
| Hong Kong | $y = 0.8402x^2 - 6.7274x + 52.046$ | 0.79 | 0.89 |
| Ukraine | $y = -0.0628x^3 + 0.7018x^2 - 2.2669x + 3.8923$ | 1.00 | 1.00 |

The source: Calculated by the authors according to the Table 1.

The functions obtained show that in the countries studied the volume of international income from tourism tends to increase in the forecast period. This is indicated by the values of the correlation coefficient, which are close to 1, i.e., the relationship between periods and revenues is linear and with each passing year the revenues increase is projected.

The most advantageous model for Ukraine is the polynomial model of the third degree (Figure 6): $y = -0.0628x^3 + 0.7018x^2 - 2.2669x + 3.8923$. The determination coefficient $R^2 = 1.0$ indicates a high degree of correlation of the dependent variables in the model. According to Fisher's criterion Fcalc > Ftable (0.75 > 0.041), the model used for the forecast should be considered adequate. Reliable forecasts will allow timely identification of possible threats to the country's economy and taking effective decisions.

Tourists visit the countries of Europe most frequently: there are six European countries among 10 countries in the rank. The region benefits due to the rich cultural heritage, developed infrastructure, and services. Having a high level of economic development, substantial experience in organizing tourism, a developed traffic network, localized historical, cultural and natural places of interest, France and Spain (Table 4 and Figure 7) are leaders by the number of travelers, while European Ukraine was visited only by 14.1 million tourists in 2018. The State Border Guard Service of Ukraine registered an increase in tourists from the non-border countries in 2018, from Europe in particular: Spain—by 68%, Great Britain—by 47.3%, Lithuania—by 23.4%, Italy—by 15.4%, Germany—by 13.3%, France—by 9.2%, India—by 57.4 %, China—by 38.8%, Japan—by 38.3%, Israel—by 21.7%, and the USA—by 19% (Ministry for Development of Economy, Trade and Agriculture of Ukraine 2018). The border movement has decreased, on the contrary. Such changes in the structure of tourism flows have resulted from the activation of the international tourism market, bilateral cooperation, visa liberalization, and increasing

proposals of direct and cheap flights in Ukraine. Ukraine has become more available for foreigners who find affordable pleasures in the form of gastronomic, wine, and event tourism there.

**Table 4.** Foreign Tourists' Arrival to the Territory within the National Borders (millions of people).

| Country | Years | | | | | | In % 2019 to 2018 |
|---|---|---|---|---|---|---|---|
| | 2014 | 2015 | 2016 | 2017 | 2018 | 2019 | |
| France | 83.7 | 84.4 | 82.6 | 86.7 | 89.3 | 89,4 | 100.1 |
| Spain | 64.9 | 68.1 | 75.3 | 81.8 | 82.7 | 83.7 | 101.2 |
| The USA | 75.3 | 77.7 | 76.4 | 77.1 | 79.7 | 79.3 | 99.5 |
| China | 55.6 | 56.8 | 59.2 | 60.7 | 62.9 | 65.7 | 104.5 |
| Italy | 48.5 | 50.7 | 52.3 | 58.2 | 61.5 | 64.5 | 104.9 |
| Turkey | 39.8 | 39.5 | 30.3 | 37.6 | 45.7 | 51.2 | 112.0 |
| Mexico | 29.3 | 32.1 | 35.1 | 39.3 | 41.3 | 45.0 | 108.9 |
| Germany | 33.0 | 35.0 | 35.5 | 37.4 | 38.9 | 39.6 | 101.8 |
| Thailand | 24.8 | 29.9 | 32.5 | 35.6 | 38.2 | 39.8 | 104.2 |
| Great Britain | 32.6 | 34.4 | 35.8 | 37.6 | 36.3 | 39.4 | 108.5 |
| Ukraine | 12.7 | 12.4 | 13.3 | 14.2 | 14.1 | 13.7 | 97.2 |

The source: Developed by the authors based on the World Tourism Barometer (2020).

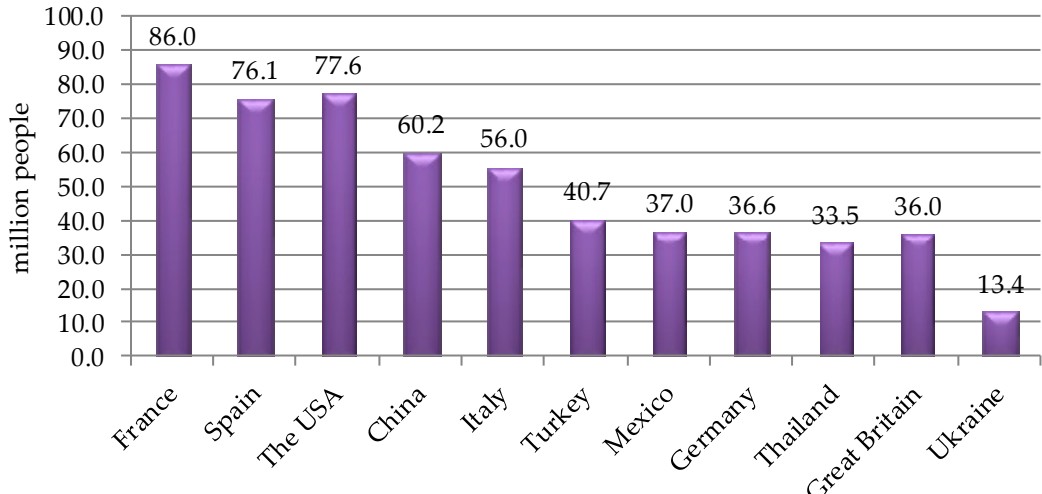

**Figure 7.** Foreign Tourists Arrival within National Borders: A Generalized Indicator for the Last 6 years (million people). The source: Developed by the authors based on the World Tourism Barometer (2020).

In 2019, 1.5 billion international tourist arrivals were registered in the world, which is 4% more than in the previous year. Thus, according to UNWTO, tourism has been "growing" for the tenth year in a row. According to the last report there was an increase in international arrivals in all regions of the world in 2019. However, uncertainty around Brexit, Thomas Cook bankruptcy, geopolitical and social tension and the global economic slowdown restrained tourism growth in 2019 in comparison to the exceptional rates of 2017 (+7%) and 2018 (+6%). The Middle East became the most rapidly growing region for international tourist arrivals in 2019 (+8%, which is almost twice as much as the previous year) (Statistics UNWTO 2020).

In 2019, Europe received 743 million international tourists (51% of the world market). America (+2%) showed an ambiguous picture, since many island directions in the Caribbean Basin revived after hurricanes in 2017 whereas arrivals to South America decreased partly because of durable social and political turmoil. Insufficient data for Africa (+4%) indicate preserving high performance in North Africa (+9%), whereas in 2019 in sub-Saharan Africa, the growth rates slowed down (+1.5%) (Statistics UNWTO 2020). At the same time the UNWTO states that the existing tendency towards

increasing international tourism flows requires responsibility in managing them, distributing them around the world in those volumes that could be managed by local communities.

The UNWTO predicted an increase in tourism flows by 3–4% in 2020. The major sport events, including the Olympic Games in Tokyo, and cultural events, such as EXPO-2020 in Dubai, were expected to have a positive effect on tourism industry. But according to the organization's forecasts the number of international tourists will fall by 20–30% when compared to the indexes of 2019 because of the COVID-19 pandemic and it will cost 5–7 years of the sector development (International Tourism and Covid-19 2020).

The Global Business Travel Association calculated that the industry could lose up to USD 820 billion because of canceled business trips, conferences, and exhibitions. Half of these expenditures may concern China. More and more companies are restricting or canceling any trips of their employees. According to the inquiry, three out of four companies have cancelled all traveling to China, Hong Kong, Taiwan, and other countries and territories of the Asia–Pacific region. Every second company has cancelled all or almost all travels to Europe. In total, 43% of company-members of the Global Business Travel Association have cancelled business travels.

The UNWTO has conducted research on the effect of tourism on the global economy in 185 countries of the world and drawn a conclusion that tourism and adjacent industries generated 10.3% of the world GDP or USD 8.9 trillion in 2019 (The World Travel and Tourism Council WTTC). Economies of many countries considerably depend on tourism and its contribution to the gross national product may be up to 80%; international tourism revenue is a main source of receiving currency (Table 5 and Figure 8).

**Table 5.** Total Contribution of Tourism Industry to the GDP of Countries (billion US$).

| Country | Years | | | | | | Contribution of Tourism Industry to the GDP in 2019, % |
|---|---|---|---|---|---|---|---|
| | 2014 | 2015 | 2016 | 2017 | 2018 | 2019 | |
| The USA | 1348.2 | 1408.0 | 1438.5 | 1540.6 | 1595.1 | 1667.7 | 7.8 |
| China | 1052.1 | 1150.6 | 1229.3 | 1351.4 | 1509.4 | 1580.8 | 11.0 |
| Japan | 316.0 | 308.8 | 352.2 | 349.1 | 367.7 | 390.9 | 7.7 |
| Germany | 331.6 | 292.3 | 299.7 | 321.4 | 344.8 | 353.1 | 9.2 |
| Great Britain | 300.8 | 299.6 | 279.2 | 288.9 | 310.9 | 323.1 | 11.4 |
| Italy | 261.9 | 233.5 | 237.7 | 253.4 | 274.9 | 279.4 | 14.0 |
| France | 270.3 | 232.0 | 229.6 | 242.2 | 265.8 | 271.4 | 10.0 |
| India | 183.8 | 193.3 | 207.0 | 233.3 | 247.3 | 277.1 | 9.6 |
| Spain | 195.1 | 168.4 | 175.4 | 192.3 | 211.0 | 221.4 | 15.9 |
| Mexico | 213.1 | 201.0 | 184.8 | 198.4 | 209.4 | 218.1 | 17.3 |
| Ukraine | 7.4 | 4.9 | 5.1 | 6.3 | 6.8 | 7.1 | 4.6 |

The source: Developed by the authors based on the World Data Atlas (n.d.).

In terms of money, the greatest contribution of tourism industry to the GDP of a country was registered in the USA and China: it makes 7.8% and 11.0%, respectively. Mexico ranks among the top ten most visited countries in the world. Tourism provides 17.3% of the gross domestic product and is one of the main sources of currency for the country. Small island countries with exotic nature depend on tourists' money most of all. For instance, tourism industries of Macau, the Seychelles and Maldives earn from 66 to 72% of the county's GDP (World Data Atlas n.d.). The share of tourism contribution to the GDP of Ukraine in 2019 made 4.6% indicating low efficiency of the tourism industry and a low level of using tourism resources.

Tourism expenditures continued to increase against the background of the global economic recession in 2014–2019 (Table 6 and Figure 9). For instance, comparing 2018 with 2017, in France an increase in international tourism expenditures among the biggest five global markets of outbound tourism made (+10.3%), whereas China (+7.5%) and the USA (+7.3%) were unquestionable leaders due to the strong currencies of these countries. In Germany, there was an increase of 6.6%, in 2018 Germans traveled 20 days on average—this is a new record for the country. Analyzing 2019, it can be

noted that there was a decrease in international tourism expenses in all countries studied due to the global economic downturn.

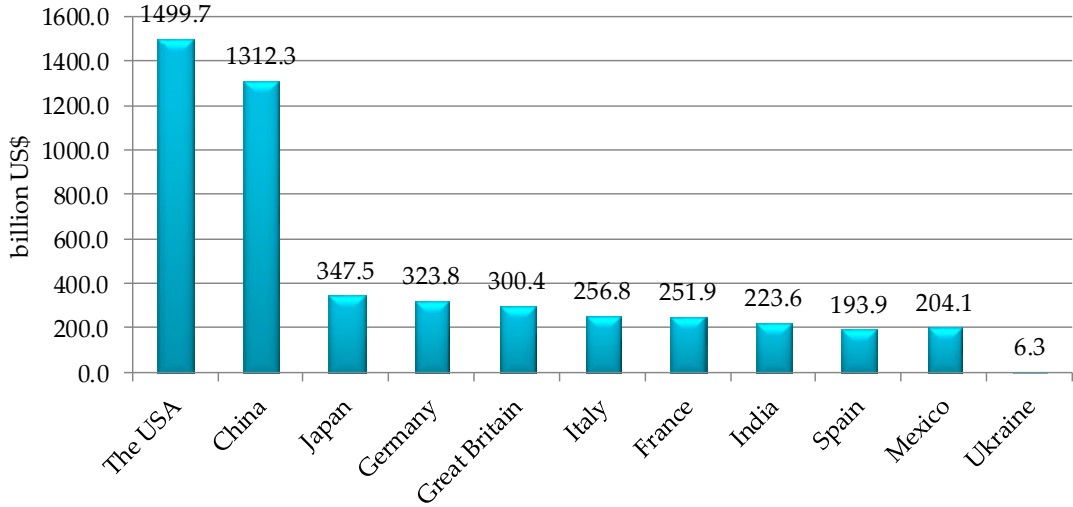

**Figure 8.** The Total Contribution of the Tourism Industry to the Country's GDP, a Generalized Figure for the Last 6 Years, billion USD. The source: Designed by the authors based on the World Data Atlas (n.d.).

**Table 6.** International Tourism Expenditures (billion US$).

| Country | Years | | | | | | Growth Rates 2019/2018 |
| --- | --- | --- | --- | --- | --- | --- | --- |
| | 2014 | 2015 | 2016 | 2017 | 2018 | 2019 | |
| China | 227.344 | 249.831 | 250.112 | 257.875 | 277.345 | 292.855 | 5.6 |
| The USA | 140.558 | 150.044 | 160.959 | 173.760 | 186.508 | 189.389 | 1.5 |
| Germany | 101.737 | 85.334 | 87.414 | 97.777 | 104.204 | 102.887 | −1.3 |
| Great Britain | 67.042 | 68.116 | 67.220 | 65.177 | 68.888 | 68.884 | 0.0 |
| France | 58.464 | 47.713 | 49.029 | 52.495 | 57.925 | 57.376 | −0.9 |
| Australia | 35.666 | 34.071 | 35.718 | 39.710 | 42.351 | 43.104 | 1.8 |
| Russia | 55.383 | 38.432 | 27.654 | 35.584 | 38.791 | 38.195 | −1.5 |
| Italy | 28.866 | 30.312 | 30.584 | 34.819 | 37.644 | 37.579 | −0.2 |
| South Korea | 26.136 | 27.957 | 29.817 | 34.453 | 34.769 | 35.371 | 3.5 |
| Canada | 34.444 | 30.271 | 28.936 | 31.811 | 33.581 | 33.307 | −0.8 |
| Ukraine | 5.470 | 5.408 | 6.306 | 7.536 | 8.287 | 8.410 | 1.5 |

The source: Designed by the authors based on the NationMaster (b).

However, in large tourism markets of the developing countries such as Brazil and Saudi Arabia, there was a reduction in tourism expenditures. In the first half of 2019, the quantity of outbound travels from China, the largest "provider" of tourists, increased by 14%, though the expenditures fell by 4%. In Ukraine, there is a considerable increase in the index of outbound tourism, visa-free travel has predictably favored it, removing many barriers for traveling around Europe. Another factor of leaving abroad is an increasing number of low-cost carriers. New players in the Ukrainian market of air transportation, the first Ukrainian low-cost carrier SkyUp, development of transport infrastructure, and the creation of new routes give more opportunities for traveling, make leisure more available and flights from regional airports more comfortable, and competition encourages development of a better tourism product for a more reasonable price.

In 2020, the UNWTO celebrates the Year of Tourism and Rural Development. This period was expected to promote the development of rural communities, creation of new jobs, stimulation of economic growth and cultural development. However, the global COVID-19 pandemic has determined other priorities for international tourism development. Global stress related to the COVID-19 pandemic has become a stimulus for innovative development of new technologies based on computerization, automatization and robotization. Since the beginning of the coronavirus pandemic robots have helped

fight against it, disinfecting surfaces, asking people to stay at home, scanning faces, cleaning floors, and delivering food. Taking into consideration the present achievements, more extraordinary machines could be expected to emerge in the future (Brouder 2020; Gössling et al. 2020; Qiu et al. 2020).

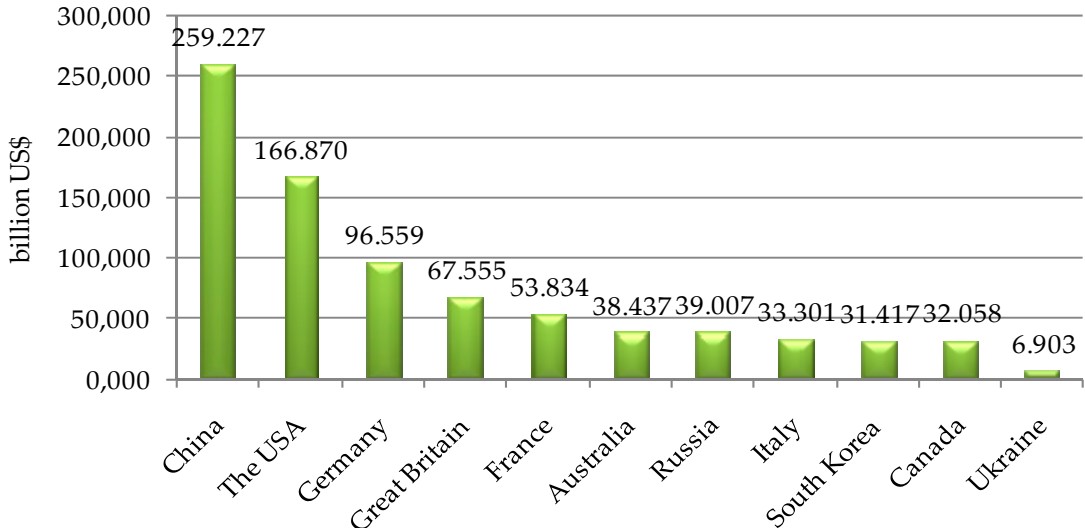

**Figure 9.** Expenditures on International Tourism, a Generalized Indicator for the Last 6 years, (billion US dollars). The source: Developed by the authors based on the NationMaster (b).

Under conditions of global risks, only a travel company that possesses modern automated information technologies that directly affect its competitiveness is able to operate effectively. The use of extensive computer systems and networks, the Internet and Internet technologies, software for all business processes of the tourism business is not just a matter of today's leadership and competitive advantage, but also the ability to create favorable conditions in the near future in the tourism market.

## 5. Discussion and Conclusions

In many countries of the world, tourism is one of the highest priority industries, which contributes significantly to the gross national income, and earnings from foreign tourism is the main source of currency and tax revenues. At present, thanks to the regional distribution of tourist flows and the beginning of mass tourist exchanges, the leadership of the world tourism industry is confidently held by Europe. This region is very popular with both locals and foreign tourists. The second position is firmly taken by America, especially North America. These two regions are key players in the global tourism market and receive the lion's share of global profits from this business.

Over the past half century, the dynamics of international tourism in other regions of the world has changed significantly with the development of relatively young Asia–Pacific, Middle East, and Africa tourist regions, which, due to negative political and economic factors, do not fully use their existing potential. Such regions include Ukraine, which, as the geographical center of Europe, is becoming increasingly interesting for foreign tourists. But in order to be competitive in the world market, it is necessary to promote the tourism product, make it better, and expand cooperation with European countries (especially neighboring countries) in order to create common tourist routes and joint promotion of tourism products.

International tourism, which for many previous years has shown a steady upward trend, has today proved to be the most vulnerable sector of the economy. The coronavirus epidemic, which is spreading around the world, has certainly affected the tourism market in many countries and made significant adjustments in travel. In 2020, the so-called microcations claim to be the main tourist trend. Tourists will realize their potential by short vacations closer to home, away from beaten tourist routes; this will save time, money, reduce the impact of travel on the environment, and provide an impetus

for the domestic tourism development. The new travel rate could mean switching to renting houses instead of hotels, switching to motor vehicles instead of flights, and increasing the use of insurance and personal travel consultants.

Today, the second wave of the pandemic is growing in the world and countries continue to fight the coronavirus disease. However, in 2002–2004 there was a similar situation when the spread of SARS in the world caused billions in losses to the tourism industry of Canada, China, Taiwan, Singapore, and the entire south-eastern region of Asia. At that time, people did not refuse to travel abroad, but simply postponed them, so even today we can predict that sooner or later the effect of deferred demand will work.

The governments around the world, especially those with a developed tourism industry, are responding to the crisis caused by COVID-19 with various policy measures. In particular, the Swiss hotel loan company SGH provides a deferral of depreciation for up to one year. The US government has introduced an incentive package of USD 2 trillion, open to all businesses, with travel coming first, while lawmakers create special piles of money for the most affected industries, including airlines, airports, and travel agencies. In Spain, the payment of interest and loans to entrepreneurs in the tourism industry has been suspended for one year and the payment of interest and/or the principal amount of loans by regions to companies and self-employed workers affected by the crisis has been postponed. In Italy, emergency allowances have been introduced for tourism and cultural workers. The social protection network has been expanded to include seasonal workers in the field of tourism and entertainment. Taxes, social insurance, and social security contributions have been withheld, and compulsory insurance contributions have been suspended. There is a refund for vouchers already provided for travel and travel packages canceled as a result of COVID-19. In France, all tourism professionals are allowed to replace the loan repayment in the equivalent amount with the next service. EUR 18 billion has been provided to the tourism sector to support its reconstruction. Support began with EUR 6.2 billion in guaranteed loans to 50,000 companies in this sector. A EUR 1.3 billion recovery plan is funded by the Caisse des Dépôts and Bpifrance. The impact of the crisis is felt throughout the tourism ecosystem, so its recovery requires a common approach and coordinated action by governments around the world, as tourism services are highly interdependent (Tourism Policy Responses to the Coronavirus (COVID-19)).

The measures introduced today by the countries' governments will shape the development of tourism in the future. It is extremely important to restore the tourism industry, while ensuring security, justice, and the absence of negative effects on the climate. It is necessary to determine the priority areas of action, and these are first of all: state policy in the field of tourism; willingness to cooperate with the private sector; assistance in implementing new protocols and effective procedures in proportion to existing risks; innovative technologies to minimize physical contact; development of new forms of domestic tourism (specialized travel, individual tourism), etc. Thanks to the coordinated, step-by-step actions of state governments, it is possible to get out of the crisis and return the industry to active growth rates.

There can be noted the historical stability of tourism, its capacity to recover quickly, and create new jobs after a global crisis is over, and that is why, in modern conditions with the spread of coronavirus, tourism enterprises face an important task to optimize risks. Our research shows that countries with a high percentage of tourism revenues in the country's GDP are the most vulnerable to changes in the global state of affairs. To prevent risks, it is advisable to forecast the income from tourism in the country's economy on the basis of correlation and regression analysis. Both correlation and simple linear regression are used to study the existence of a linear relationship between two variables, providing certain assumptions about the data. However, the results of the analysis need to be interpreted with caution, especially when looking for a cause-and-effect relationship or when using a regression equation for prediction (Bewick et al. 2003). Regression determines how x causes y to change, and the results will change if x and y are swapped. Correlated x and y are variables that can be swapped to get the same result. Correlation is a separate statistic or data point, while regression

is a whole equation with all data points represented by a line. Correlation shows the relationship between two variables, while regression allows seeing how one affects the other. The data shown with regression establish the causes and consequences, when one changes, the other changes as well, and not always in one direction; with correlation, the variables move together (Calvello 2020).

The rapid development of information technology will provide more reliable and faster methods of processing, transmitting, receiving and storing all types of information needed to effectively address the main challenges of the tourism business and international cooperation, especially during and after the COVID-19 pandemic. Thus, to work out marketing strategies for developing the industry and attracting potential tourists, it is advisable to use existing social media. They are an ideal tool for finding information to plan travel and allow the tourist to make decisions in a simple and easy way. The bottom line is that the traveler shares a post related to an interesting tourist attraction, a route, as well as his experience in this journey, through social media, which attracts the attention and interest of other members of social networks (Cheunkamon et al. 2020).

Prospects for further research will be fueled by the need to manage the information support of economic entities in the field of international tourism to reduce risks.

**Author Contributions:** Conceptualization, Y.K., V.H., V.B., A.K. and L.B.; Data curation, Y.K., V.H., V.B., A.K. and L.B.; Formal analysis, Y.K., V.H., V.B., A.K. and L.B.; Funding acquisition, Y.K., V.H., V.B., A.K. and L.B.; Investigation, Y.K., V.H., V.B., A.K. and L.B.; Methodology, Y.K., V.H., V.B., A.K. and L.B.; Project administration, Y.K., V.H., V.B., A.K. and L.B.; Resources, Y.K., V.H., V.B., A.K. and L.B.; Software, Y.K., V.H., V.B., A.K. and L.B.; Supervision, Y.K., V.H., V.B., A.K. and L.B. All authors have read and agreed to the published version of the manuscript.

**Funding:** This research received no external funding.

**Acknowledgments:** The authors are very grateful to the anonymous referees for their helpful comments and constructive suggestions.

**Conflicts of Interest:** The authors declare no conflict of interest.

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
