# Peer review of "International Tourism Development in the Context of Increasing Globalization Risks: On the Example of Ukraine’s Integration into the Global Tourism Industry"

_jrfm, doi:10.3390/jrfm13120303_

Round 1
Reviewer 1 Report
Abstract and Title
I would suggest the author(s) simplifying the title and the abstract.
The abstract should have the following virtual instructions: background, methodology, main results, theoretical or practical implications.
Introduction
As reported here, the introduction is not scientific now. The background of the study must have and must be supported by scientific citations.
Additionally, I would suggest the author(s) include the following information in this section:
- background of the study (almost done but it needs scientific support);
- GAP of the study, why it is important for your study considering the main literature?
- Brief method explanation;
- Brief theoretical and practical implications obtained with your study.
- A map of the paper.
Literature review
The literature review is well-written and organized.
Materials and Methods
Now the methodology is far from scientific. I don't understand if it is a professional methodology from UNWTO or scientific. Please, clarify it also considering previous studies in the literature which use this method. Otherwise, please explain that it is a new method.
Additionally, the authors should also explain what they used for their analysis in terms of statistical software.
Results
The results are interesting. However, due to the lack of the method could be difficult to understand it. Please, clarify all the steps that you perform and also the results could be more clear.
Conclusions
This section is almost clear. However, I suggest the following point:
- be more linear with the literature that you cite in the literature review section. Please provide the main theoretical implications of your study.
All the best!
The reviewer
Author Response
Dear Reviewer,
Thank you very much for your time and valuable comments.
A Point-by-Point Response to Reviewer 1 is attached.
Best regards,
The authors

Reviewer 2 Report
Dear Authors,
Currently, the tourism sector is an interesting object of scientific research. Therefore, the submitted manuscript follows this trend. However, However, I have a lot of comments about this paper.
Detailed comments:
- Introduction:
- no research gap specified;
- the main goal is too complex;
- no references to literature (e.g. line 54);
- in general, the introduction poorly defines the research problem;
2. Literature Review
-
literature review should be improved (there are many positions by Ukrainian authors);
- no detailed review of the research methods used in other articles;
3. Materials and Methods:
- no justification for the choice of period;
- what is the substantive justification for the choice of Ukraine?
- what is the justification for the selection of research method;
4.Results:
- no references to literature (e.g. line 183);
- the models in Figures 1-6 are out of date and not suitable for forecasts
(COVID-19 pandemic will significantly reduce tourism revenues); - table 2 is redundant - a repetition of figures 1-6;
- most of the tables and graphs are only present the information in a given period;
- moreover, each country has a different area, so comparing the number of tourists or revenues numerically is wrong;
5. Missing chapter on discussions.
Reviewer
Author Response
Dear Reviewer,
Thank you very much for your time and valuable comments.
A Point-by-Point Response to Reviewer 2 is attached.
Best regards,
The authors

Reviewer 3 Report
Dear authors,
Thank you for participating in the study of such an interesting topic as: «International Tourism Development in the Context of Increasing Globalization Risks».
After reading the article, I´d like to do some suggestions that can improve its quality in my opinion.
Some references are not correctly cited, for example, (UNWTO Statistics), (World Data Atlas), (Ranking of countries in the world by population in 1980-2024), (Ministry for Development of Economy, Trade and Agriculture of Ukraine).
It is necessary to read the text more carefully and not to do repetitions (р.4). «Summarizing the scholars’ views, we can state that international tourism is a form of international economic relations, affecting a considerable number of economic sectors, consisting of many social components and aimed at improving the countries’ well-being.».
Self-citation (Boiko et al. 2019). (Boiko 2016) is not correct, the article is about other things that do not relate to tourism (p. 6). You should review the self-citation in the article. When referring to works, it is advisable to cite a specific page to resolve any misunderstandings.
Analysis of the results of tables and figures is somewhat confusing, I would suggest you to review it. It is appropriate to add, for example, incorrect analysis of table 2. «The highest rates of income from international tourism among the countries studied in Hong Kong, the country receives from 5.1 to 6.5 thousand dollars per capita». In fact, there was a drop from 6.5 in 2014 to 5.1 in 2018.
The writing of the article should be improved. There is no text transition from the USA, Thailand to Ukraine (p.5). Incorrect text transition (p.11) regarding covid-19 risk and transition to 2017 text and data. Once again on covid (p 14). The text should be reformatted. The logic of presenting the material is lost.
Digital data is given without indication of time periods. (p. 13). For instance, in France an increase in international tourism expenditures among the biggest five global markets of outbound tourism made (+10.3%)...
There is also outdated information. «News of a possible new outbreak of the virus in the fall scares away tourists who cancel their travel plans». The de facto outbreak occurred.
The conclusions are too short, you could include which is your contribution to the academic knowledge. There are no generalizations of your data analysis in the conclusions. The expediency of the given quotations in the conclusions is not clear: Risks can be predicted…. (Bewick et al. 2003), Regression determines…. (Calvello 2020), Thus, to work out… (Cheunkamon et el. 2020).
In general, the article needs improvement regarding its presentation and clarification.
Kind Regards

Author Response
Dear Reviewer,
Thank you so much for your time and valuable comments.
A point-by-point response is attached.
Best wishes,
The authors

Round 2
Reviewer 1 Report
Dear Author(s),
Thank you for your improvements.
All the best.
The reviewer
Author Response
Dear Reviewer,
Thank you very much for your time and valuable comments.
Best regards,
The authors
Reviewer 2 Report
Dear Authors,
I regret to say that the corrections introduced by you in the article are insufficient.
My new comments:
- New title. The new title does not fully reflect the content of the article.
After changing the title, I don't know why pay special attention to Ukraine, if it is not a leading tourist country. It is difficult for me to find a substantive justification for this. - Main purpose. "The main purpose of the article is to analyze current trends and determine the prospects for international tourism in the world in the face of increasing globalization risks". There is no answer to the main purpose of the work - Where is the trends? Where did you identify prospects? 2018 is a thing of the past, not a trend or perspectives. The UNWTO is already identifying the first prospects.
- Methods. I didn't notice any significant changes. Regression models can be used for forecasts, but it is already known that substituting data in the model will not give reliable results for 2020.The models identified in the work may only be useful for determining the 2019 revenues. Therefore, there is no need for these models. If you believe that these models are important, then it is worth making forecasts based on them, e.g. until 2025. Specialists' calculations are already available that the tourism industry will not return to the state of 2019 so quickly. Maybe it is better to focus on only one country (e.g. Ukraine) and show what are the tourism trends in that country in the context of increasing globalization risks.
- The chapter on discussion should be significantly expanded.
Author Response
Dear Reviewer,
Thank you so much for your time and valuable comments.
A point-by-point response is attached.
Best regards,
The authors
